# Correlations between Crawl Kinematics and Speed with Morphologic, Functional, and Anaerobic Parameters in Competitive Swimmers

**DOI:** 10.3390/ijerph19084595

**Published:** 2022-04-11

**Authors:** Marek Strzała, Arkadiusz Stanula, Piotr Krężałek, Kamil Sokołowski, Łukasz Wądrzyk, Marcin Maciejczyk, Jakub Karpiński, Wojciech Rejdych, Robert Wilk, Wojciech Sadowski

**Affiliations:** 1Department of Water Sports, Faculty of Physical Education and Sport, University of Physical Education, 31-571 Kraków, Poland; marek.strzala@awf.krakow.pl (M.S.); kamil.sokolowski@awf.krakow.pl (K.S.); 2Institute of Sport Sciences, Jerzy Kukuczka Academy of Physical Education, 40-065 Katowice, Poland; j.karpinski@awf.katowice.pl (J.K.); w.rejdych@awf.katowice.pl (W.R.); r.wilk@awf.katowice.pl (R.W.); w.sadowski@awf.katowice.pl (W.S.); 3Department of Physiotherapy, Faculty of Motor Rehabilitation, University of Physical Education, 31-571 Kraków, Poland; piotr.krezalek@awf.krakow.pl; 4Department of Biomechanics, Institute of Biomedical Sciences, University of Physical Education, 31-571 Kraków, Poland; wadrzyk504@gmail.com; 5Department of Physiology and Biochemistry, Faculty of Physical Education and Sport, University of Physical Education, 31-571 Kraków, Poland; marcin.maciejczyk@awf.krakow.pl

**Keywords:** hand surface area, range of motion, arm-cranking, countermovement jump

## Abstract

The purpose of this study was to examine the relationship between a unique complex of predictors and 100 m front crawl race kinematics and swimming speed. In 28 male competitive swimmers (age: 19.6 ± 2.59 years), the following groups of predictors were assessed: (a) the morphologic, (b) the functional upper limb range of motion, and (c) the anaerobic indices of arm-cranking and a series of countermovement jumps. The Pearson product-moment correlation coefficient was calculated to distinguish the predictors and the swimming results. The main finding was that the indices of the power (arm-cranking) and the work (countermovement jump) generated in the anaerobic tests showed a significant and higher correlation with stroke length and stroke index than total body length, upper limb range of motion, or hand and forearm surface area. These results were obtained in accordance with the high swimming economy index relation to clear surface swimming speed. This study reveals that the strength generated by the limbs may represent a predictor of swimming kinematics in a 100 m front crawl performance.

## 1. Introduction

In competitive swimming, the selected best athletes, who have passed the subsequent stages of selection and training, have a unique compilation of morphological features. This applies to the height and weight of the body and, at the same time, the linearity of the body which is adequate to achieve the hydrodynamic traits to swim fast [1]. In systematic training, given the variability and plasticity of human morphology, some features are formed into the expected shape [2,3,4].

Many years of observations have led to the understanding that elite competitive swimmers need an inherited potential for the development of functional features, e.g., for the front crawl, a large range of motion of the upper limbs in the sagittal plane is needed; this is related mainly to their length, but also to the range of motion in the shoulder joints [5,6,7,8]. It can also be seen that the size of the body segments involved in generating propulsion in water is of particular importance. For the anticipated advantage in front crawl swimming, the surface area of the hand and forearm has been researched [9,10,11]; what is equally significant is that the muscle area of the body parts that perform propulsive movements has been considered [12,13,14].

Competitors who train systematically in the water make a constant compromise between producing the propulsion force and the size, shape, or structure of the body, which creates resistance during body motion. The body’s slenderness, surface structure, and alignment become more important as speed increases [15,16]. The swimmer’s drag increases at least quadratically as speed rises, thereby affecting the energy expenditure. Efforts are thus being made to minimize the drag forces as much as possible [17,18]; on the other hand, training work to increase muscle strength and endurance remains essential to overcome the given level of drag forces. This generally applies to the force generated by the upper limbs [13,19], as well as by the lower limbs [20,21].

In general, in swimming, the somatic and functional strength and endurance properties translate into the swimming effect, which can be measured by kinematic indicators. Indeed, the calculation of these spatiotemporal indices, such as stroke rate (*SR*), stroke length (*SL*), and stroke index (*SI*)—the swimming economy index—is used in a large number of scientific studies [22], but it is also routinely used, especially with regard to *SR*, in the control of current training.

The aim of the present study was to reveal the relationship between 100 m front crawl race kinematics and (a) morphologic indices such as the surface area of the hand (*A*_h_) or the hand and forearm (*A*_hf_), total body length (*TBL*), body mass (*BM*), and the body mass index (*BMI*); (b) functional properties, such as the front crawl imitation range of motion of the upper limb (*ROM*_A_), elbow (*ROM*_E_), and dactylion (*ROM*_D_); and (c) the anaerobic muscle motor system capability indices of the arms (arm-cranking) and legs (a series of countermovement jumps).

At the same time, the influence of these mentioned predictors on the speed of swimming in a group of mixed-skill male competitive swimmers was examined. We assumed that the influence of body size and the functional cranio-caudal upper limb range of motion, as well as general anaerobic strength and power, would have a similar significant influence on the kinematics and swimming speed.

## 2. Methods

### 2.1. Participants

A cohort study was designed to analyse the predictors enabling the swimmers to achieve different levels of swimming results. A total of 28 male volunteers participated in this observational research. All of them were healthy and had licences from the Polish Swimming Federation. They were educated at the high school (9 swimmers) and university level (19 swimmers). Their mean age, height, *BM*, and *BMI* were as follows: 19.6 ± 2.59 years (range: 17–25 years), 183.8 ± 6.32 cm, 77.7 ± 9.60 kg, and 23.09 ± 1.75 kg · m^–2^, respectively. *TBL* was measured when lying back, from the tips of the fingers (with arms stretched up above the head) down to the pointed toes (foot plantar flexion).

Of the 28 swimmers, 4 participated in regional meets, 20 in national meets, and 4 were experienced in international tournaments. Their personal best results at these 3 levels of competition in 100 m freestyle were: 79.4%, 87.7%, and 92.6%, respectively (expressed as a percentage of the world record). The swimmers participated in 2 training sessions a day, 6 days a week. The study was approved on 19 April 2018 by the University Ethics Committee at the Jerzy Kukuczka Academy of Physical Education in Katowice, which consented to the conduct of the research as confirmed by Resolution No. 8/2018.

### 2.2. Estimation of the Hand and Forearm Surface Area

The propulsion surface area of selected body segments was estimated by using the photographic method with a DSC-RX100IV camera (Sony, Japan). The tested upper limb segments of the swimmer were placed flat on a table, with a bend at the elbow closest to a right angle. Next to the limb, on the dark background and visible in the camera frame, was a yellow square with a known area. The camera with which the photographs were taken was suspended on a rigid frame at a fixed distance from the tabletop, so that its optical axis was perpendicular to its surface. The high-resolution photographs obtained in this way (5472 px × 3648 px) were then processed in the MS Paint software; the areas of the tested segments in each photograph were covered with colour (Figure 1).

The photos prepared in this way were then analysed in the ImageJ software (version 1.52a, National Institutes of Health, Bethesda, MD, USA; http://rsbweb.nih.gov/ij/, accessed on 30 January 2020). The number of pixels contained in each photo in the area of the analysed segment and in a square with a known area was calculated with the *Color_Pixel_Counter* plugin. By comparing these two results, an estimate was obtained of the propulsion surface area of the segment, being its projection onto the tabletop surface. The forearm was separated by drawing a straight line from the top of the elbow (A) to the innermost incision on the opposite side (B) (Figure 1); after separating the segment, *A*_hf_ (cm^2^) was calculated. To separate the hand from the forearm, a straight line was drawn from the styloid process of the ulna (C) to the styloid process of the radius (D). This made it possible to separate the hand and calculate *A*_h_ (cm^2^). The reliability of the measurement was as follows: the intraclass correlation coefficient (ICC)_(2,1)_ = 0.983 (95% CI: 0.932–0.996) for *A*_hf_, and the ICC_(2,1)_ = 0.971 (95% CI: 0.887–0.993) for *A*_h_.

### 2.3. Determining the Ranges of Motion of Selected Anatomical Points in the Imitation of Swimming Movement

Measurements were carried out by using the three-dimensional motion analysis system SMART-D (BTS, Milano, Italy). The system consisted of six synchronized cameras (70 Hz) recording infrared light reflected from passive markers placed on the upper limb of the examined person at three anatomical points: the acromion, the lateral epicondyle of the humerus, and the dactylion. Markers were attached to the subject’s body at the selected anatomical points by one experienced researcher, a physiotherapist. While the examined subject standing in an upright position was performing a movement that imitated the front crawl swimming technique, the displacement of markers placed on his body was registered. The obtained results were then processed in the BTS system modules: Tracker and Analyzer. The performed calculations allowed the values of the vertical (cranio-caudal) marker displacements, which are an estimate of the horizontal linear ranges of the motion of individual body parts in swimming pool conditions, to be obtained. Figure 2 shows the tracks of the markers and linear ranges of motion during one movement cycle.

### 2.4. Anaerobic Power of Arms

An arm-cranking test was conducted with the use of modified apparatus (834E-Ergomedic, Monark, Sweden). The load of the competitor performing the test was set adequately to his body weight at the level of 4.5% of *BM* [23]. The average power (*P*_cra ave_) (W) and maximal power (*P*_cra max_) (W) expended during a period of 20 s physical exercise were measured; they were also measured in relation to *BM*: *P*_cra ave rel_ (W · kg^–1^) and *P*_cra max rel_ (W · kg^–1^). A 4 min warm-up of continuous arm cranking, with a cadence of 90 min^–1^ at a minimal workload, was applied prior to the maximal exhaustive effort. Five-second sprints were cranked against a higher workload at the cadence of ca. 90–100 min^–1^, first in the middle and then close to the end of the warm-up [24].

### 2.5. Countermovement Jump Test (CMJ)

A series of 10 countermovement jumps was performed on a force plate (AMTI BP 400600, Lynchburg, VA, USA), rigidly fixed to the floor. The frequency of jumps—1 jump every 2 s—was imposed by using a metronome. At the beginning of the test, the athlete stood upright on the platform with his weight evenly distributed between both feet. Hands were placed on the hips throughout the whole test to eliminate their contribution to energy generation. The work generated in a single jump (*W*_CMJ_) (J), averaged over the 10 jumps, was taken as an absolute indicator of anaerobic muscle system motor capabilities. The average elevation of the centre of mass (*h*) (cm) in the 10 jumps was considered as an indicator of muscle system motor capabilities relative to *BM*.

The CMJ test was performed using a platform (AMTI BP 400600, USA) mounted in the laboratory floor. The subject started the test standing still on the platform, trying to evenly distribute the weight of the body over both feet. The athletes then performed a series of 10 CMJ jumps at a frequency imposed on them by the metronome (one jump every 2 s). In order to eliminate the participation of the upper limbs in the test performed, the hands of the participants rested on their hips throughout the entire measurement. The work performed during a single jump (*W*_CMJ_) (J) was taken as the absolute indicator of the motor abilities of the muscular system, while the maximum elevation of the centre of body mass in a single jump (*h*) (m) was used as a relative indicator of muscle system motor capabilities (in relation to *BM*). Both indicators were averaged for 10 jumps.

The test was preceded by a dynamic warm-up, and its procedure was previously applied by Mitchel et al. [21]. It included dynamic stretching, shuttle runs with progressive speed, and also body-weight squats.

### 2.6. The 100 m Front Crawl Race

A race in a short-course pool with the use of an automatic timing device (Omega, Switzerland) was conducted as in a competition repechage but with the use of in-water starts.

The kinematic variable *SR* (cycle · min^–1^) was assessed using pictures captured by camcorders (JVC GC-PX100BE, Japan). The *SR* was assessed using three following strokes of the middle section of each lap of the race. The *SL* calculation was based on the data gathered in 9 m sectors. For the purpose of this measurement, another JVC GC-PX100BE camcorder (all 4 cameras were working at the sampling rate of 100 fps) was used. The camera recordings were synchronized with a flashlight. The gathered recordings provided an average time of swimming the 9 m sectors (Δ*t*), allowing the calculation of the 36 m surface swimming speed (*V*_surface_). The *SL* kinematic was a product of *V*_surface_ and *SR*: *SL* = *V*/*SR* (m).

The swimming time in the other start, turn, and finish zones was used to calculate the *V*_STF_ of 64 m, and to receive the *V*_total100_, the whole race time was taken.

The video picture recordings and the arrangement of the cameras, followed by picture analysis and the calculation of the basic kinematics, were performed analogically to the previous one [25], but in this study, a swimming distance twice as long was taken into account. 

The availability of the above-mentioned kinematic indices made it possible to calculate the *SI* (m^2^ · cycle · s^-–1^), which was calculated as *V*_surface_ · *SL*. The ICC value for the *SR* calculation process equalled 0.99 (95% CI: 0.960–0.997).

### 2.7. Statistical Analysis

The normal Gaussian distribution of the data was verified by the Shapiro–Wilk test. The Student’s *t*-test for dependent samples was applied for normally distributed data and equal variances. The magnitudes of the differences between the results of the swimming speed (*V*_surface_ vs. *V*_total100_) were expressed as standardized mean differences (Cohen effect sizes [ES]). The criteria to interpret the magnitude of the ES values were as follows: <0.2, trivial; 0.2–0.6, small; 0.6–1.2, moderate; 1.2–2.0, large; and >2.0, very large. The Pearson product-moment correlation coefficient was calculated to assess relationships: (1) between *V*_surface_ and front crawl stroke kinematics and (2) between stroke kinematics or 100 m front crawl results (*V*_surface_, *V*_total100_) and (a) morphological, (b) upper limb range of motion, and (c) anaerobic indices of arms and legs. The statistical significance was set at *p* ≤ 0.05. All statistical analyses were conducted by using the Statistica 13.3 software (TIBCO Software Inc., Palo Alto, CA, USA).

## 3. Results

The two separated average speeds (*V*_surface_, *V*_total100_) of the 100 m front crawl race were significantly different (*p* ≤ 0.001, ES = 1.09 [moderate], 95% CI: 0.90–1.36) (Figure 3).

The spatiotemporal *SI* index was significantly and highly related to *V*_surface_, but the *SL* kinematics correlated moderately and non-significantly with *V*_surface_ (Table 1).

Selected morphological indices showed a relationship with the stroke kinematics; *BM* was significantly related to *SI*, while the relationship of *BMI* with the *SL* and *SI* stroke kinematics was close to significant (Table 2).

There was a non-significant relationship between the range-of-motion indices of *ROM*_A_, *ROM*_E_, and *ROM*_D_, respectively, and the *SL* and *SI* kinematics (Table 3).

There were significant relationships between the arm-cranking anaerobic indices *P*_cra ave_ and *P*_cra max_ and the *SL* and *SI* stroke kinematics (Table 4).

The *CMJ* test–*W*_CMJ_ index was significantly related to the *SL* and *SI* kinematics (Table 5.)

In Table 6, among the groups of morphologic, arm and leg anaerobic capabilities, and the range-of-motion indices, only the best predictors for the 100 m front crawl performance are presented.

## 4. Discussion

In this study, the main finding is that the indices of the power (arm-cranking) and work (countermovement jump) generated in the anaerobic tests showed a significant correlation and a higher relationship with *SL* and *SI* than *TBL*, upper limb range of motion, or *A*_hf_. These results were obtained in accordance with the *SI* relation to *V*_surface_. Thus, the study reveals that the strength generated by the limbs remarkably implies the shape of the swimming kinematics in the 100 m front crawl performance.

Training toward the desired level of skills or, in other words, obtaining the desired level of kinematic indicators, translates into the swimming speed achieved [26]. It is also likely that swimmers are able to achieve the desired speed with a different combination of *SR* and *SL*, which, in turn, can be modelled by other factors. Nevertheless, a regularity is noticeable in those results, i.e., a larger body and a more adequate arm span [27] of the swimmer translates into a bigger *SL* [28]. Moreover, a longer body, the *TBL* index in our study, is crucial; as considered by Kjendlie and Stallman [29], it is quite evident that tall swimmers will have a higher hull speed and a potentially lower wave-making resistance compared with shorter swimmers. The authors also reported that elite swimmers at maximal sprinting speed exhibit a higher Froude number than younger swimmers in a developmental phase, and at any given submaximal speed, a taller swimmer will present a lower Froude number and thus lower wave-making resistance. The swimming kinematic indices noted in our study were not significantly influenced by body length or the surface area of the most important propulsion segments of the limbs (*A*_h_ and *A*_hf_). Kjendlie and Stallman [29] state that the propelling surface of a large hand area is beneficial in swimming and makes for a higher propelling efficiency. This is also in accordance with a study by Toussaint et al. [30], in which increasing the propelling size by paddles raised the propelling efficiency by 8% compared with swimming with hands only and caused an *SL* increase. Gourgoulis et al. [31] also reported that hand paddles increased the propelling efficiency, *SL*, and the swimming velocity, mainly because of the larger propulsive areas of the hand in comparison with free swimming. They noted a significant decrease in *SR* and suggested that it might argue for the effectiveness of hand paddle training, particularly when large paddles are used in front crawl swimming. However, in competitive swimming, the generation of propulsion force may be more dependent on how the stroke is performed. Marinho et al. [32] demonstrated that slightly spread fingers of the propelling hand could allow the creation more propulsive force, and a more perpendicular attack angle of hand models showed the highest values for the drag coefficient. Samson et al. [33] analysed a swimmer’s hand–forearm model with visualizations of the flow and flow vortex structure and reported that the hand force coefficients were approximately twice as large as those of the forearm. They noted that the total force coefficients were highest for angles of attack between 40° and 60°, and, further, at the same angle of attack, the forces produced when the leading edge was the thumb side were slightly greater than those produced when the leading edge was the little finger side. Nevertheless, such characteristics can be seen in the ‘S-shaped’ front crawl hand-stroke pattern [34].

In this study, an even better relationship was observed between the overall anaerobic motor ability of the arm and leg muscles; the somatic features indicate that in the highly trained competitive swimmers, these properties may be predominant [35,36]. In our diverse sample of competitive swimmers, we could expect to see a greater impact of the hand’s range of motion or A_hf_ on the race results. For these swimmers, however, the better results of the race were determined by anaerobic capacity potential, where *V*_total_ was related to *P*_cra max_ (0.37, *p* < 0.05) and to *W*_CMJ_ (0.40, *p* < 0.05). Training for strength or power in swimming was discussed in former studies [24,37]; today, overall strength and endurance are also part of the swimmer’s multi-factor racing preparation [23].

Hand and forearm size may already play a secondary role in shaping stroke kinematics and speed because in adult swimmers it does not vary much between individuals of the same body size [38,39]. In swimming, there is a well-established knowledge of the so-called high elbow and the importance of already obtaining a high angle of attack of the hand and forearm during the early phase of the pull [40]. This ability requires both great arm and shoulder flexibility and muscle endurance [41]. Deficiency in the flexibility and strength of the shoulder girdle may result in a less optimal trajectory of the hand and forearm, a so-called low elbow stroke path of arm, lower generation of propulsion force [42], or lower *SL* [6]. In our study, in the group of swimmers, who were diverse in terms of morphology and training and who also prefer their own competitive distances, the 100 m front crawl performance (*V*_total_) was related to *TBL* (r = 0.43, *p* < 0.05), functional hand range of motion (*ROM*_D_) (r = 0.39, *p* < 0.05), and *BM* (r = 0.39, *p* < 0.05). This relationship is common among competitive swimmers, bearing in mind the body height differences between the peers or short-, middle-, and long-distance front crawl swimmers [1,43]. In our study, *TBL* or *ROM*_A_, *ROM*_E_, and *ROM*_D_ were not significantly related to *SL* or *SI*, but a swimmer with a longer body possesses an additional advantage during pool swimming, for which a swimmer with a shorter one is not able to compensate by strength and conditioning. It is likely that the taller swimmer’s centre of mass is higher on the block at the beginning of race and that this produces a longer start, i.e., an entry farther from the block. During lap swimming in a pool (such as in our study), a longer swimmer’s body is beneficial: the centre of mass does not travel the whole race distance owing to its centre of mass turning and finishing at a longer distance from the pool wall [29].

These observations show that the shape of the kinematics in sprint swimming is not as strongly correlated with body size and the surface of the propulsion segment of the arms as could be assumed. Thus, the dominant role was played by the power and work generated in the anaerobic tests. The limitation of the findings is the use of general anaerobic tests, which can be supplemented by tests focused on the specificity of the swimming stroke.

## 5. Conclusions

The main finding was that the indices of the power (arm-cranking) and work (countermovement jump) generated in the anaerobic tests showed a significant and higher correlation with stroke length and stroke index than total body length, upper limb range of motion, or hand and forearm surface area. These results were obtained in accordance with the high swimming economy index relation to clear surface swimming speed. This study reveals that the strength generated by the limbs is the parameter that showed the highest correlation with swimming kinematics in the 100 m front crawl performance.

## Figures and Tables

**Figure 1 ijerph-19-04595-f001:**
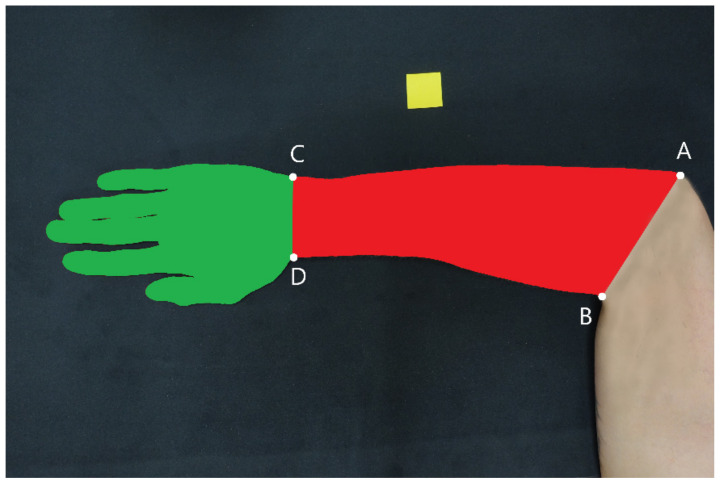
A photograph showing the examined segments of the upper limb with the applied colour and a square with a known surface area. Lines of separation: from the top of the elbow (A) to the innermost incision on the opposite side (B) and from the styloid process of the ulna (C) to the styloid process of the radius (D).

**Figure 2 ijerph-19-04595-f002:**
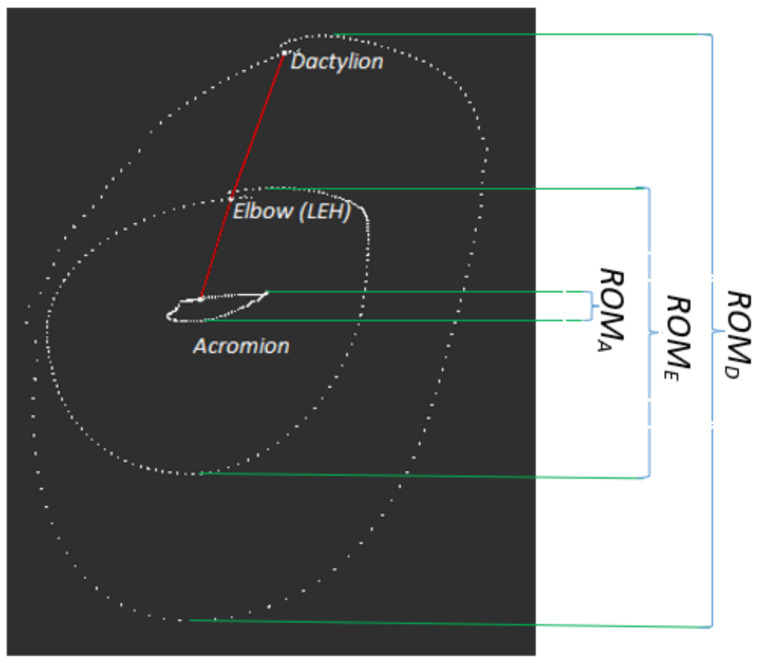
Tracks of markers placed on the subject’s body and linear ranges of motion (ROM) in a single cycle of movement imitating crawl swimming (side view).

**Figure 3 ijerph-19-04595-f003:**
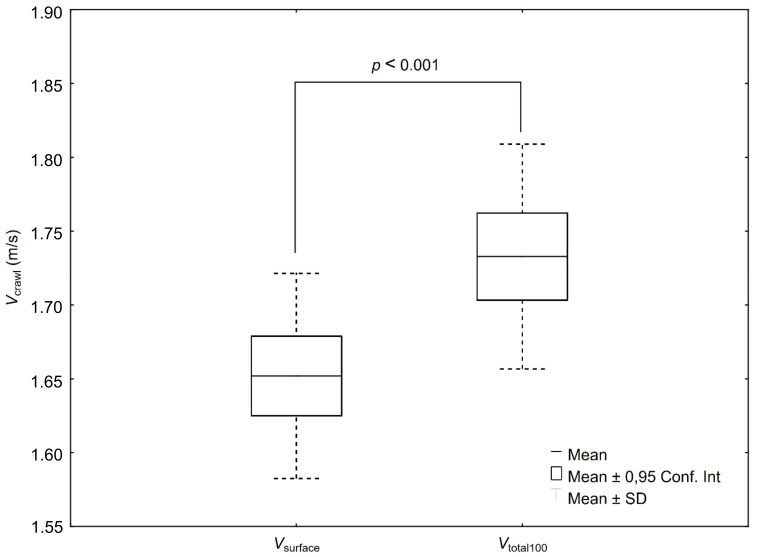
Swimming speed in the 100 m front crawl race (*V*_total100_) and the separated speed of swimming on the surface (*V*_surface_).

**Table 1 ijerph-19-04595-t001:** Values of *SR*, *SL*, *SI* of 100 m crawl and correlation with *V*_surface_.

Parameter	*SR* (cycle · min^–1^) 46.84 ± 3.98	*SL* (m) 2.14 ± 0.19	$\mathrm{SI}\;(\frac{m^{2}}{s})$ 3.53 ± 0.38
*V*_surface_ (m · s^–1^)	0.14	0.30	0.64 ***

*** significant relationship between the analysed indices with *p* < 0.001.

**Table 2 ijerph-19-04595-t002:** Values of *BM*, *BMI*, *TBL*, *A*_h_, *A*_hf_, and correlation with stroke kinematics.

Parameter	*BM* (kg) 77.72 ± 9.60	*BMI* (kg · m^–2^) 23.09 ± 1.75	*TBL* (cm) 253.80 ± 9.77	*A*_h_ (cm^2^) 159.89 ± 15.49	*A*_hf_ (cm^2^) 425.13 ± 40.57
*SR*	–0.22	–0.30	–0.10	–0.18	–0.22
*SL*	0.32	0.35 (*p* = 0.07)	0.22	0.23	0.27
*SI*	0.37 *	0.35 (*p* = 0.07)	0.31	0.25	0.29

* significant relationship between the analysed indices with *p* < 0.05.

**Table 3 ijerph-19-04595-t003:** Values of longitudinal, cranio-caudal axis upper limb range-of-motion indices and correlation with stroke kinematics.

Parameter	*ROM*_A_ (m) 0.12 ± 0.001	*ROM*_E_ (m) 0.73 ± 0.001	*ROM*_D_ (m) 1.38 ± 0.01
*SR*	–0.03	–0.09	–0.03
*SL*	0.07	0.15	0.17
*SI*	0.08	0.18	0.29

**Table 4 ijerph-19-04595-t004:** Values of arm-cranking anaerobic strength indices and correlation with stroke kinematics.

Parameter	*P*_cra ave_ (W) 452.78 ± 82.06	*P*_cra__ave rel_ (W · kg^–1^) 5.71 ± 0.60	*P*_cra__max_ (W) 507.96 ± 99.01	*P*_cra__max rel_ (W · kg^–1^) 6.38 ± 0.73
*SR*	–0.32	–0.28	–0.31	–0.28
*SL*	0.39 *	0.30	0.40 *	0.33
*SI*	0.41 *	0.26	0.43 *	0.32

* significant relationship between the analysed indices with *p* < 0.05.

**Table 5 ijerph-19-04595-t005:** The *CMJ* anaerobic muscle motor system capability indices: *h*, *W*_CMJ_, and correlation with stroke kinematics.

Parameter	*h* (m) 0.38 ± 0.04	*W*_CMJ_ (J) 287.44 ± 51.72
*SR*	–0.31	–0.34
*SL*	0.34	0.43 *
*SI*	0.32	0.47 *

* significant relationship between the analysed indices with *p* < 0.05.

**Table 6 ijerph-19-04595-t006:** Significant predictors of 100 m front crawl performance (*V*_surface_, *V*_total100_), selected from groups of morphologic, anaerobic strength, and range-of-motion indices.

Parameter	*BM*	*TBL*	*P* _cra_ _max_	*W* _CMJ_	*ROM* _D_
*V* _surface_	0.28	0.30	0.27	0.29	0.28
*V* _total100_	0.39 *	0.43 *	0.37 *	0.40 *	0.39 *

* significant relationship between the analysed indices with *p* < 0.05.

## Data Availability

Not applicable.

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
