# Peer review of "Correlations between Crawl Kinematics and Speed with Morphologic, Functional, and Anaerobic Parameters in Competitive Swimmers"

_ijerph, 2022, doi:10.3390/ijerph19084595_

Round 1
Reviewer 1 Report
Dear authors,
I would like to congratulate you for the choice of a very interesting topic, however, in the introduction of the article it is not very clear what the objective of this research is, as it would perhaps be more useful to carry out the study with non-professional swimmers, in search of future talents.
In addition, I would like the authors to mention in this article, other published research (Specific and Holistic Predictors of Sprint Front Crawl Swimming Performance, DOI: https://doi.org/10.2478/hukin-2021-0058) and very similar to the current one, and to know if the sample has been the same, as plagiarism has been detected in some paragraphs of the manuscript (arm cracking test, countermovement jump test and swimming race).
I suggest the authors to modify this paragraphs and to clarify the main aim of the investigation for usefulness in talent selection.
Author Response
Dear Sir/Madam,
Thank you for taking the time to read the manuscript of our research paper in-depth and kindly write down valuable suggestions on how to improve it. We carefully read all the comments and made the necessary corrections to the best of our current capabilities. We would like to apologize if not all general comments have been received and processed into fresh content for our manuscript. Nevertheless, we welcomed all specific comments and made appropriate changes.
Comments/Suggestions 1:
I would like to congratulate you for the choice of a very interesting topic, however, in the introduction of the article it is not very clear what the objective of this research is, as it would perhaps be more useful to carry out the study with non-professional swimmers, in search of future talents.
Response 1:
Thank you again for the positive perception of our work.
Comments/Questions/Suggestions 2
I would like the authors to mention in this article, other published research (Specific and Holistic Predictors of Sprint Front Crawl Swimming Performance, DOI: https://doi.org/10.2478/hukin-2021-0058) and very similar to the current one, and to know if the sample has been the same, as plagiarism has been detected in some paragraphs of the manuscript (arm cracking test, countermovement jump test and swimming race).
Response 2:
At that time, we repeated the well-established anaerobic tests twice on a similar sample of swimmers as a reference to the newly designed water-specific measurements and the new morphological-functional measurements incorporated in to this manuscript. It looks like we ran out of time (before the end of February) to make appropriate changes to the description of the methods used, for which we apologize. Thus, we have introduced the necessary changes and we have provided an appropriate source of literature.
Comments/Suggestions 3:
I suggest the authors to modify this paragraphs and to clarify the main aim of the investigation for usefulness in talent selection.
Response 3:
Thank you, of course we have made a correction.

Reviewer 2 Report
The idea of this study is interesting. My recommendations are as follows:
I recommend extending the introductory section with new relevant aspects focused on the age category of the subjects of this study.
I recommend that all data in the tables be mentioned to two decimal places.
I recommend that in the results section an interpretation of the obtained data be made.
Author Response
Dear Sir/Madam,
Thank you for taking the time to read the manuscript of our research paper in-depth and kindly write down valuable suggestions on how to improve it. We carefully read all the comments and made the necessary corrections to the best of our current capabilities. We would like to apologize if not all general comments have been received and processed into fresh content for our manuscript. Nevertheless, we welcomed all specific comments and made appropriate changes.
Comments/Suggestions 1:
The idea of this study is interesting. My recommendations are as follows: I recommend extending the introductory section with new relevant aspects focused on the age category of the subjects of this study.
Response 1:
Thank you for the suggestion, we've made one change to the introductory part. Currently, in this short period, we have not managed to introduce more thorough changes in this chapter.
Comments/Questions/Suggestions 2
I recommend that all data in the tables be mentioned to two decimal places.
Response 2:
We made changes to the tables taking into account the reviewer's suggestion
Comments/Suggestions 3:
I recommend that in the results section an interpretation of the obtained data be made.
Response 3:
We have introduced some changes related to table 2 data.

Reviewer 3 Report
Thank you for the opportunity to review an interesting topic „The relation of 100-m front crawl kinematics and speed to morphologic, functional, and anaerobic predictors in competitive swimmers“
The manuscript was written in a correct manner and I recommend that the paper be aaccepted.
I have only one comment that relates to the directions of further investigations. Maybe the authors have set them up.
The main question of this study was to examine the relationship between a unique complex of predictors (morphologic, functional of upper limb range of motion, anaerobic indices of arm-cranking and series of countermovement jumps and 100-m front crawl race kinematics and swimming speed).
I consider this topic relevant in the sports science field as the results generated by authors enable them to be applied in practice and better prepare high-performance (elite) athletes for the International Championships and/or Olympics. More specifically, on the basis of the paper results, the structure of microcycles, the nature and specificity of sports training can be optimised. In the aftermath of all this, good results in in swimming sport and medals at the International Championships can be expected.
I believe that the authors' study complements material published by other authors and is an integral element of the didactic of athletes' workout. In other words, according to the specific data of the manuscript, it is possible to model various volumes, intensities and directions during sports exercises in the micro-cycle programme, all of which form the basis for the development of the aerobic and anaerobic power of high-performance swimmers.
The methodology is well designed and described. Nevertheless, the authors could indicate the type of study carried out (experiment or observational study; if the observational study was fixed, then what the subtype: cross-sectional, cohort or case-control study).
Of course, the approval I have given is positive in this respect. The authors responded fully to a question from the study. That's enough. From the point of view of sports science, a more specific purification of empirical and practical applicability of the results generated by authors would certainly be welcome.
Table 3 numbers “0.12 ± 0.00” and “0.73 ± 0.00” could be adjusted.
Best Regards
Author Response
Dear Sir/Madam,
Thank you for taking the time to read the manuscript of our research paper in-depth and kindly write down valuable suggestions on how to improve it. We carefully read all the comments and made the necessary corrections to the best of our current capabilities. We would like to apologize if not all general comments have been received and processed into fresh content for our manuscript. Nevertheless, we welcomed all specific comments and made appropriate changes.
Comments/Suggestions 1:
The manuscript was written in a correct manner and I recommend that the paper be aaccepted.
Response 1:
Thank you again for the positive perception of our work.
Comments/Questions/Suggestions 2
I have only one comment that relates to the directions of further investigations. Maybe the authors have set them up.
The main question of this study was to examine the relationship between a unique complex of predictors (morphologic, functional of upper limb range of motion, anaerobic indices of arm-cranking and series of countermovement jumps and 100-m front crawl race kinematics and swimming speed).
I consider this topic relevant in the sports science field as the results generated by authors enable them to be applied in practice and better prepare high-performance (elite) athletes for the International Championships and/or Olympics. More specifically, on the basis of the paper results, the structure of microcycles, the nature and specificity of sports training can be optimised. In the aftermath of all this, good results in in swimming sport and medals at the International Championships can be expected.
I believe that the authors' study complements material published by other authors and is an integral element of the didactic of athletes' workout. In other words, according to the specific data of the manuscript, it is possible to model various volumes, intensities and directions during sports exercises in the micro-cycle programme, all of which form the basis for the development of the aerobic and anaerobic power of high-performance swimmers.
The methodology is well designed and described. Nevertheless, the authors could indicate the type of study carried out (experiment or observational study; if the observational study was fixed, then what the subtype: cross-sectional, cohort or case-control study).
Response 2:
We would like to thank you for your valuable insights and comments. Yes, we agree, our observations among swimmers should provide this environment with indications that, assessing the state of strength, endurance and the shaping of the kinematics of the strokes, should be applicable in modifying the training procedure. We participate, co-create this swimming community and try to be helpful.
Comments/Suggestions 3:
The methodology is well designed and described. Nevertheless, the authors could indicate the type of study carried out (experiment or observational study; if the observational study was fixed, then what the subtype: cross-sectional, cohort or case-control study).
Response 3:
Thank you, we added it.
Comments/Suggestions 4:
Of course, the approval I have given is positive in this respect. The authors responded fully to a question from the study. That's enough. From the point of view of sports science, a more specific purification of empirical and practical applicability of the results generated by authors would certainly be welcome.
Response 4:
Yes, we almost always talk to the coaches and consider introducing our results into the current and long-term development of swimmers.
Comments/Suggestions 5:
Table 3 numbers “0.12 ± 0.00” and “0.73 ± 0.00” could be adjusted.
Response 5:
Thank you for paying attention, we have adjusted it

Round 2
Reviewer 1 Report
I appreciate the changes authors have made, however, they didnt specify where along the manuscript.
I still detect some plagiarism (34%).
I strongly recommend to correct this.
Thank you
Author Response
Dear Reviewer,
Thank you again for your time and useful suggestions.
We have corrected the selected fragments of the manuscript in order to obtain satisfactory changes.
All new contents we marked by red fonts and we have marked all modifications by crossing out the words, and additionally, we have highlighted these deletions in yellow.
Kind regards,
Authors
Best Regards